# The Use of microRNAs in the Management of Endometrial Cancer: A Meta-Analysis

**DOI:** 10.3390/cancers11060832

**Published:** 2019-06-16

**Authors:** Romain Delangle, Tiphaine De Foucher, Annette K. Larsen, Michèle Sabbah, Henri Azaïs, Sofiane Bendifallah, Emile Daraï, Marcos Ballester, Céline Mehats, Catherine Uzan, Geoffroy Canlorbe

**Affiliations:** 1Cancer Biology and Therapeutics, Centre de Recherche Saint-Antoine (CRSA), Sorbonne University, INSERM UMR_S_938, 75020 Paris, France; annette.larsen@mfex.com (A.K.L.); michele.sabbah@inserm.fr (M.S.); sofiane.bendifallah@aphp.fr (S.B.); emile.darai@aphp.fr (E.D.); catherine.uzan@aphp.fr (C.U.); 2Assistance Publique des Hôpitaux de Paris (AP-HP), Department of Gynecological and Breast Surgery and Oncology, Pitié-Salpêtrière University Hospital, 75013 Paris, France; henri.azais@aphp.fr; 3Assistance Publique des Hôpitaux de Paris (AP-HP), Department of Obstetrics and Gynaecology, Tenon University Hospital, 75020 Paris, France; tiphainefc@hotmail.com; 4Centre National de la Recherche Scientifique (CNRS), 75012 Paris, France; 5Department of Gynecology, Groupe Hospitalier Diaconesses Croix Saint-Simon, 75020 Paris, France; MBallester@hopital-dcss.org; 6INSERM U1016-Institut Cochin, UMR 8104, Team “From Gametes to Birth”, University Paris Descartes, 75014 Paris, France; celine.mehats@inserm.fr; 7Institut Universitaire de Cancérologie (IUC), 75020 Paris, France

**Keywords:** endometrial cancer, microRNA, survival, nodal involvement, prognostic

## Abstract

*Introduction*: Endometrial cancer (EC) is the most important gynecological cancer in terms of incidence. microRNAs (miRs), which are post-transcriptional regulators implicated in a variety of cellular functions including carcinogenesis, are particularly attractive candidates as biomarkers. Indeed, several studies have shown that the miR expression pattern appears to be associated with prognostic factors in EC. Our objective is to review the current knowledge of the role of miRs in carcinogenesis and tumor progression and their association with the prognosis of endometrial cancer. *Materials and Method*: We performed a literature search for miR expression in EC using MEDLINE, PubMed (the Internet portal of the National Library of Medicine) and The Cochrane Library, Cochrane databases “Cochrane Reviews” and “Clinical Trials” using the following keywords: microRNA, endometrial cancer, prognosis, diagnosis, lymph node, survival, plasma, FFPE (formalin-fixed, paraffin-embedded). The miRs were classified and presented according to their expression levels in cancer tissue in relation to different prognostic factors. *Results*: Data were collected from 74 original articles and 8 literature reviews which described the expression levels of 261 miRs in ECs, including 133 onco-miRs, 110 miR onco-suppressors, and 18 miRs with discordant functions. The review identified 30 articles studying the expression pattern of miR in neoplastic endometrial tissue compared to benign and/or hyperplastic tissues, 12 articles detailing the expression profile of miRs as a function of lymph node status, and 14 articles that detailed the expression pattern of miRs in endometrial tumor tissue according to overall survival or in the absence of recurrence. *Conclusions*: The findings presented here suggest that miR analysis merits a role as a prognostic factor in the management of patients with endometrial cancer.

## 1. Introduction

Endometrial cancer (EC) is the most important gynecological cancer in terms of incidence with 380,000 new cases being diagnosed each year worldwide [1]. The most common histological type is endometrioid adenocarcinoma [2]. The current classification of endometrioid adenocarcinoma is based solely on histology which directs the subsequent therapeutic management [3]. However, this approach is often insufficient for prognosis. Therefore, the inclusion of additional criteria such as biological expression profiling will be needed to adapt both the surgical management and adjuvant therapy.

The molecular classification of endometrial cancer has revealed an important heterogeneity of tumors with comparable histological type and grade. Four molecular subtypes have been proposed, including (1) microsatellite instability hyper-mutated, (2) copy-number-low microsatellite stable, (3) copy-number-high serous-like, and (4) DNA polymerase epsilon (*POLE*) ultra-mutated. Histologically, this last group corresponds to endometrioid tumors [4].

Ongoing research aims to identify and characterize novel biomarkers for better understanding of the different subtypes, to provide improved assessment of prognosis, and for optimization of patient care. microRNAs (miRs) are particularly attractive candidates as biomarkers. miRs are post-transcriptional regulators that are implicated in a variety of cellular functions including carcinogenesis [5,6] and resistance to treatment [7]. Typically, miRs display an RNA sequence complementary to the messenger RNA (mRNA) of a given gene, resulting in targeting and degradation of this mRNA, thereby leading to selective transcriptional repression [8]. However, a single miR may target several mRNAs and have either oncogenic or tumor suppressor activity depending on the tissue in which it is expressed [5,6]. Importantly, next-generation sequencing techniques can detect and quantify [9] the expression of miR in both fresh and paraffin-embedded tissues [10], as well as from liquid biopsies [11].

Regarding endometrial cancer, several studies have shown that the miR expression pattern appears to be associated with prognostic factors such as lymph node involvement [12,13,14], lymphovascular space invasion (LVSI) [15], overall survival [14,16,17,18,19,20,21,22,23,24,25,26,27] (OS), and recurrence-free survival (RFS) [13,14,25,26].

Improved knowledge of the expression profile of miRs would likely explain certain molecular mechanisms associated with EC and could serve as a basis for diagnostic tests, prognosis, or the identification of potential novel therapeutic targets. To the best of our knowledge, there is no comprehensive recent review of the expression of miRs in EC. The objective of this review is to review the current knowledge of the role of miRNAs in carcinogenesis and tumor progression and their association with the prognosis of endometrial cancer.

## 2. Materials and Methods

A literature search for miR expression in EC was carried out using the following databases:

MEDLINE, PubMed (the Internet portal of the National Library of Medicine), http://www.ncbi.nlm.nih.gov/sites/entrez?db=pubmed.

The Cochrane Library, Cochrane databases “Cochrane Reviews” and “Clinical Trials”. http://www3.interscience.wiley.com/cgi-bin/mrwhome/106568753/HOME DARE.

We used the following terms: microRNA, endometrial cancer, prognosis, diagnosis, lymph node involvement, survival, plasma, FFPE (formalin-fixed, paraffin-embedded).

The database search was further supplemented by using original articles, reviews, and meta-analyses, including the work cited within. Only articles published in English or French were included. The literature used was published between 1 November 2008 and 31 October 2018.

The miRs were classified and presented according to their expression levels in cancer tissue in relation to different prognostic factors. The expression profiles of the miRs according to lymph node status and survival were also explored. Finally, the relationship between the selected miRs in the plasma or serum and the presence of endometrial cancer is presented.

## 3. Results

Data were collected from 75 original articles and 8 literature reviews published between 1 November 2008 and 31 October 2018 which described the expression levels of 261 miRs in ECs, including 133 onco-miRs, 110 miR onco-suppressors, and 18 miRs with discordant functions (miR-18b, 23a*, -31, -34a, -130b, -142-3p, -142-5p, -146a, -184, -185, -194, -200c, -202, -204, -326, -432, -630, -760) (Appendix A).

### 3.1. Expression Profile of miRs Associated with Malignant Endometrial Tissues Compared to Healthy or Hyperplastic Endometrial Tissues

Our literature review identified 30 articles studying the expression pattern of miR in neoplastic endometrial tissue compared to in benign and/or hyperplastic tissues [13,14,23,24,25,28,29,30,31,32,33,34,35,36,37,38,39,40,41,42,43,44,45,46,47,48,49,50,51,52]. The studies included an average of 45 endometrial tumor samples (minimum 7, maximum 141), with well-defined histological types for 15 studies. The studies included an average of 18 healthy endometrial samples (minimum 5, maximum 48). The healthy endometrium was derived from operative specimens of patients operated on for benign pathologies, except for nine studies in which the healthy endometrium was taken from an area adjacent to the tumor.

miR extraction was carried out from paraffin or frozen tissue. For determination of the relative expression levels, on-chip hybridization techniques were used in ten studies and next-generation sequencing (NGS) was used in two studies, all validated by RT-qPCR. Compared to heathy endometrial tissues, endometrial tumors showed the following: 

Increased expression of the following miRs: miR-9, -9*, -9-3p, -10a, -18a-3p, -19b, -25-5p, -27a, -31, -34a, -95, -96, -103, -106a, -106b, -107, -130b, -135a, -135b, -141, -142-5p, -146, -146b-5p, -151, -153, -155, -181a, 181c-3p, -181c, -182, -183, -184, -191, -193-3p, -194, -200a, -200a*, -200a-5p, -200b, -200b*, -200c, -203, -205, -210, -215, -221, -223, -218, -301, -325, -326, -330, -337, -363, -423, -425, -429, -432, -449, -499, -518d-5p, -520c-5p, -522, -526a, -1202, -5787, and -6749-5p.

Decreased expression of the following miRs: miR-10b, -10b* -21, -23a*, -29c, -30a-3p, -30a-5p, -30c, -31, -32, -33b, -99a, -99a-3p, -99b, -100, -101, -126, -127-3p, -133b, -139-5p, -152, -185, -193, -193a, -193b, -195, -196a, -196a-5p, -199b, -199b-3p, -199b-5p, -204, -214, -216b, -221, -302a-5p, -328-3p, -337-3p, -338-3p, -367-3p, -368, -369, -370, -376a, -376c, -377, -377-5p, -381, -409, -410, -411, -424, -424*, -424-3p, -431, -432, -449a, -451, -487b, -496, -503, -516, -542-3p, -542-5p, -596, -610, -630, -632, -652, -758, -760, and -1247.

A summary of these data is provided in Table 1.

### 3.2. Expression Profile of miRs According to Lymph Node Status

Our literature review identified 12 articles detailing the expression profile of miRs as a function of lymph node status [13,14,19,21,24,25,26,41,48,53,54,55].

The studies included an average of 11 samples with positive lymph node status (minimum 2, maximum 29) and 42 samples with negative lymph node status (minimum 13, maximum 121). Eight studies included only histological types. 

Positive lymph node status was associated with increased expression of miR-10a, -10b, -26a, -26a1, -34a, -95, -123, -125b1, -125b2, -133a, -143, -145a, -181a, -200a*, -203, -222-3p, and -429.

Positive lymph node status was also associated with decreased expression of miR -24b-5p, 34c-3p, -34c-5p, -184, -204-5p, and 375. 

A summary of these data is provided in Appendix A.

### 3.3. Expression Profile of miRs According to Survival

Our literature review identified 14 articles that detailed the expression pattern of miRs in endometrial tumor tissue according to overall survival or in the absence of recurrence [13,14,16,17,18,19,20,21,22,23,24,25,26,27]. Five studies included only endometrioid carcinoma [14,18,20,24,27]. RNA extraction was predominantly carried out from paraffin-preserved tissue. 

Significant improvement in overall survival was associated with the following: 

Increased expression of miR-10b*, -29b, -100, -101, -129-2, -130b, -139-5p, -152, -183-5p, -194, -199a-5p, -202, and -455-5p; 

Decreased expression of miR-200c, -205, -429, and -1228 and of the combined expression of six miRs (miR-15a, miR-142-3p, hsa-miR-142-5P, miR-3170, miR-1976, miR-146a).

Significant improvement in recurrence-free survival was associated with the following: 

Increased expression of miR-29b, -152, -199a-5p, and -455-5p;

Decreased expression of miR-429 and -1228. 

A summary of these data is indicated in Table 2.

### 3.4. Relationship between Specific miRs in the Plasma/Serum and the Presence of Endometrial Cancer

Six studies compared the expression of circulating miRs (six with plasma and one with serum) in patients with EC compared to healthy patients [11,13,14,24,56,57]. The 19 miRs miR-15b, -27a, -92a, -99a, -100, -135b, -141, -143, -186, -199b, -200a, -203, -204, -205, -222, -223, -449a, -1228, and miR-1290 showed increased expression in EC patients. The 10 miRs miR-9, -21, -30a-3p, -204, -301b, -1179, -3145-5p, -4502, -4638-3p, and -4665-5p showed decreased expression in EC patients.

A summary of these data is indicated in Appendix A.

## 4. Discussion

The findings presented here suggest that miR analysis merits a role in the management of patients with endometrial cancer, particularly when associated with prognostic factors such as lymph node status, LVSI, and recurrence-free survival; it can thereby complement the classical anatomo-pathological approach, even if there has been no integration of the miRs into either the anatomical classification or the molecular classification until now [2]. Various studies have focused on miRNAs’ implication in endometrial cancer mechanisms [58] with no fully established conclusions. Yet, based on new knowledge of miRNAs associated with different prognoses, new pathogenetic classifications for EC may be proposed including miRNAs as one element among others (i.e., anatomic prognostic features, molecular classification) in order to give better targeting for future treatment. The miRs most frequently implicated in endometrial cancer are miR-182, miR-183, miR-200a, miR-200b, and miR-205, which are overexpressed in tumor tissues, and miR-152, which is underexpressed. However, the identification of these miRs was obtained from a variety of different studies focusing on different aspects.

The data presented here are consistent with studies of the role of miRs in other cancer types. Overexpression of miR-182 was associated with a poorer prognosis in terms of overall survival (Hazard Ratio (HR) = 2.50, 95% CI: 1.86–3.36) and recurrence-free survival (HR = 2.52, 95% CI: 1.67–3.79) in a meta-analysis that collected data from patients with different tumor types [58]. Likewise, overexpression of miR-183 seems to be associated with poor prognosis in terms of overall survival (HR = 2.642, 95% CI: 2.152–3.245) and with tumor progression (HR = 2.403, 95% CI: 1.267–4.559) according to a meta-analysis by Zhang et al. [59].

Mechanistic studies have revealed a role for many miRs in carcinogenic pathways. This is particularly the case for miR 200a, 200b [60], and 205a [61], which are involved in the PI3K/Akt/mTOR signaling pathway, most likely through downregulation of the Phosphatase and TENsin homolog (PTEN) tumor suppressor. However, it has been shown that there is a high prevalence of PTEN gene mutation (56–57%) [62,63], PI3K gene mutation (36–39%) [62,63], or even the associated mutation of these two genes (25%) [63]. The loss of PTEN induces a lack of control of A phosphorylation, while the mutation of the gene coding for the p85 subunit of PI3K is involved in the overactivation of Akt, both resulting in cell proliferation and resistance to apoptosis [64]. This signaling pathway is also a therapeutic target under study; the use of a PI3K/mTOR inhibitor and the use of temsirolimus as an mTOR inhibitor appears to be effective on cells presenting these mutations [65]. Otherwise, overexpression of miR-182 inhibits the expression of cullin-5 (CUL5), the overexpression of which leads to a reduction in cell proliferation and the CUL5–RING E3 (i.e., Really Interesting New Gene E3) ligase complex [66]. miR-183 has a role in the induction of the epithelio–mesenchymal transition, in apoptosis inhibition, and in the promotion of cell proliferation by downregulating cytoplasmic polyadenylation element-binding protein 1 (CPEB1) [67]. Finally, the subexpression of miR-152 induces a decrease in the expression of PTEN, a tumor suppressor gene having a fundamental role in the inhibition of cell proliferation, particularly in EC [68].

One of the crucial difficulties in the management of patients with EC is to establish a guideline for when to perform lymphadenectomy associated with hysterectomy in the initial surgical management. To date, no imaging technique [69,70] or preoperative histological analysis [70,71,72] makes it possible to accurately define the risk of ganglionic invasion. A determination of miR expression in the primitive tumor tissue might provide some guidelines. Unfortunately, most studies that have examined the correlation between miR expression and lymph node status in EC [13,14,19,21,24,25,26,41,48,53,54,55] have included several histological types and grades, with data from secondary analyses. However, expression levels of miR-34 and miR-184 may be useful for adapting patient care, since Canlorbe et al. [53] showed that a decrease in miR-34c-5p and miR-184 was associated with positive lymph node status in patients with otherwise good prognosis.

Interestingly, decreased expression of miR-34 was also identified as a negative prognostic factor for a different type of EC, since decreased expression of miR-34 in serous-type endometrial cancer is strongly associated with LVSI [26]. This data strengthens the knowledge about the miR-34 family (miR-34a, b, and c) which seems to act as a tumor suppressor miR in many cancers [73]. Interestingly, miR-34 is a direct target of tumor protein 53 (TP53)/p53, a tumor suppressor gene resulting in cell cycle arrest and apoptosis when activated under cellular stress [74]. Many studies have previously demonstrated a dysregulation of miR-34 in several cancers, including melanoma, hepatocellular, mesothelial, colic, nasopharyngeal, leukemia [75], prostate cancer [76], neuroblastoma [77], glioblastoma [78], and breast cancer [79]. However, to date, few studies have addressed its role in EC, except for a functional study by Li et al. [80] demonstrating that miR-34c acts as a tumor suppressor miR in human endometrial cancer 1b (HEC-1b) with the E2F transcription factor 3 (E2F3) being one of the targets. It has also been reported that decreased expression of miR-184 is associated with recurrence in endometrial cancer [81].

An attractive feature of miRs is that they are relatively stable in serum, suggesting that liquid biopsies might be useful for miR analysis in different clinical situations. Several studies have used miR analysis to identify individuals with EC. The expression of miR-27a in association with cancer antigen 125 (CA-125) may be able to distinguish between healthy subjects and patients with endometrioid adenocarcinoma with an area under the curve of 0.894 (95% CI, 0.807–0.980, a sensitivity of 0.78, and a specificity of 0.97) [82]. Another study concluded that serum expression levels of four miRs (miR-222, -223, -186, and -204) allow us to distinguish between patients with endometrioid adenocarcinoma and healthy subjects with an area under the curve of 0.927 (95% CI 0.85–1.00, a sensitivity of 91.7%, and a specificity of 87.5%) [11]. According to the same study, the diagnostic performance of these four miRs was much higher than that of the classical serum marker CA-125, which has an area under the curve of 0.673 (95% CI 0.525–0.821) [11]. Although these findings are encouraging, there are limits to the use of miRs. Indeed, the reproducibility of their detection and quantification is relatively low. This is due to several factors. The accuracy, reproducibility, sensitivity, and specificity of the main PCR kits were compared by Mestdagh et al. [83], and they present a great disparity in performance. There is also currently no consensus regarding the use of housekeeping genes as endogenous internal control when assessing miRNA expression. The housekeeping genes are conventionally selected from a panel of 12 non-coding RNAs (RNU48, RNU44, U75, RNU6B, U6, U54, RNU38B, U18, U49, miR-26b, miR-92a, and miR-16) described as stable in tissues or at least used in EC studies [49,84]. Their expression nevertheless shows considerable variability according to some authors [13] and could lead to errors of interpretation and non-reproducibility of the results according to the histological types concerned.

A new technique for the detection and quantification of miRs using isothermally amplified time-gated Förster resonance energy transfer (TG-FRET) that has shown interest in breast and ovarian cancer could overcome these drawbacks in EC [85].

Regarding the analysis of miR expression in plasma and serum, we found that among the 33 miRs identified, only miRNA-223 had been described three times before [11,56,82], while miRs -186 and -222 had been described twice before [11,56]. The lack of reproducibility may be explained by two main reasons: the heterogeneity of the patient samples and the diversity of the detection techniques. The use of liquid-based cytology could be a non-invasive alternative technique to the collection of plasma samples to highlight the dysregulation of some miRs as shown by Kottaridi et al. [86]. The different studies compared subjects who were thought to be in good health with EC patients. However, the latest data in the literature clearly demonstrated that within the same histological type (here, endometrioid), there can be several molecular subtypes which display diverse mutations of genes such as PTEN, KRAS (i.e., Kirsten rat sarcoma viral oncogene homolog), β-catenin, or microsatellite instability [2], which would lead to altered miR synthesis.

Furthermore, there is no consensus on the techniques for measuring and comparing expression levels of miRs in plasma or serum. Panels of highly expressed miRs in plasma, such as miR-93, miR-26b, miR-192, miR-103a, miR-142-3p, miR-92a, miR-638, miR-16, and miR-451, may serve as “household” miRs [87,88,89,90] that can be used for standardization of other miRs. For example, in one study the crude results were normalized with five miRs—miR-93, miR-26b, miR-192, miR-103a, and miR-142-3p [24]. The same team also proposed to normalize the expression of the miRs of interest with that of *Caenorhabditis elegans* oligonucleotides that had been added to the reaction mixture, including cel-miR-39, cel-miR-54, and cel-miR-238 [14,24]. Wang et al. also performed the normalization of studied miRs with cel-miR-39 [60]. Two other studies determined the absolute concentrations of miRs using calibration curves created with known concentrations of synthetic miRs (10^−6^ fM/l) [11,13].

Recently, several reviews have focused on the potential therapeutic use of targeting miRs in cancer [7,91], including EC [55]. These reviews underlined that global suppression of miRs is not compatible with survival, since deletion of the Dicer complex, which is needed for a common step in miR biosynthesis, is lethal [92]. Any therapeutic strategy based on their use can therefore be conceived only in a targeted way directed toward one or a few miRs.

Upregulation of a suppressor miR would require an agonist-type strategy with reintroduction of the miR of interest or a functionally similar analogue. Conversely, reducing the level of an onco-miR would need an antagonistic strategy by oligonucleotides complementary to the miR in question, hence the name “antagomir” [93] or “AMO” (anti-miR oligonucleotide) [94]. Several reviews have addressed this problem with very similar conclusions [95,96,97,98]. Whatever the strategy used, it faces the same two major obstacles: the difficulty of getting these highly negatively charged molecules to penetrate the cells and their stability with respect to the nucleases present in the blood and in the cells. These are the same difficulties encountered by the use of therapeutic siRNAs (interfering RNAs). The panoply of possible chemical modifications used to overcome these problems is detailed in a review by Saumet et al. [99]. Finally, the possibility of introducing these oligonucleotide sequences by gene therapy vectors is also being actively explored.

## 5. Conclusions

miR analysis of endometrial tumor tissue complements the classical anatomo-pathological approach adding prognostic and therapeutic value. In particular, certain miR expression profiles are associated with important prognostic factors such as lymph node status and the presence of emboli. miR analysis may be simplified for routine clinical use, provided that data collection and analysis techniques are standardized.

## Figures and Tables

**Table 1 cancers-11-00832-t001:** Differences in the expression profile of microRNAs (miRs) between malignant endometrial tissue and healthy endometrial tissue.

Reference	Sample Type	Case Sample	Control Sample	miR Increased (Case vs. Control)	miR Decreased (Case vs. Control)	Detection Technique
Liu Y. et al., 2018 [28]	-	Endometrioid endometrial cancers (*n* = 30): 15 FIGO IA, 15 FIGO Ib	Adjacent healthy endometrial tissue (*n* = 30)	-	miR-101	RT-qPCR
Liu J. et al., 2018 [29]		Endometrial cancers (*n* = 25)	Endometrial tissue of healthy cases (*n* = 15)	-	miR-139-5p	RT-qPCR
Ma J. et al., 2018 [30]	Fresh Tissue, Paraffin	Endometrial cancers (*n* = 80)	Endometrial tissue of healthy cases (*n* = 56)		miR-302a-5p miR-367-3p	RT-QPCR
Huang et al., 2018 [31]	−80 °C	Endometrial cancers (*n* = 20)	Endometrial tissue of healthy cases (*n* = 20)	miR-106b	-	Array RT-qPCR
Fang et al., 2018 [51]	-	Endometrial cancers (*n* = 69): 33 N+ 36 N−	Endometrial tissue of healthy cases (*n* = 10)	miR-182, miR-183, miR-153, miR-27a, miR-96	-	RT-qPCR
Ushakov et al., 2018 [32]	-	Endometrioid endometrial cancers FIGO I-II (*n* = 32)	Adjacent healthy endometrial tissue (*n* = 32)	-	miR-29c, miR-31, miR-185, miR-652	RT-qPCR
Xie et al., 2017 [33]	Paraffin	Endometrioid endometrial cancers (*n* = 30): 12 FIGO I, 7 FIGO II, 11 FIGO III 15 Grade 1, 13 Grade 2, 2 Grade 3	Adjacent healthy endometrial tissue (*n* = 30)	-	miR-216b	RT-qPCR
Zhang S. et al., 2017 [34]	Paraffin	Endometrial cancers (*n* = 37): 21 FIGO I, 5 FIGO II, 4 FIGO III 5 FIGO IV‘ 25 Grade 1, 7 Grade 2, 5 Grade 3	Endometrial tissue of healthy cases (*n* = 22)	-	miR-101	RT-qPCR
Chen et al., 2017 [35]	−80 °C	Endometrial cancers (*n* = 15)	Hyperplasic endometrial tissue (*n* = 15) Endometrial tissue of healthy cases (*n* = 15)	miR-5787, -6749-5p, -1202	miR-338-3p, miR-449a, miR-196a	Array RT-qPCR
He et al., 2017 [36]	Paraffin −80 °C	Endometrial cancers (*n* = 68): 54 endometrioid, 14 others 55 FIGO I–II, 13 FIGO III–IV 50 Grade 1–2, 18 Grade 3 59 N+, 9 N−	Endometrial tissue of healthy cases (*n* = 20)	miR-944	-	RT-qPCR
Wang Z. et al., 2017 [37]	-	Endometrial cancers	Endometrial tissue of healthy cases (*n* = 15)	miR-522, miR-139-3p, miR-520c-5p, miR-518d-5p, miR-146b-5p, miR-34a, miR-526a, miR-193a-3p, miR-221, miR-4674	miR-760	Array RT-qPCR
Cai et al., 2016 [38]	−80 °C	Endometrial cancers (*n* = 24)	Adjacent healthy endometrial tissue (*n* = 24)	miR-337	-	RT-qPCR
Zhao et al., 2016 [39]	−70 °C	Endometrial cancers (*n* = 11)	Adjacent healthy endometrial tissue (*n* = 11)	-	miR-126	RT-qPCR
Yoneyama et al., 2015 [40]	Fresh Tissue	Endometrioid endometrial cancers I (*n* = 7): IA Grade 1–2, IB Grade 1–3, IIIA Grade 1, IIIC Grade 2	Adjacent healthy endometrial tissue (*n* = 7)	miR-200a, -200b, -429	-	Array RT-qPCR
He et al., 2015 [41]	Paraffin	Endometrioid endometrial cancers (*n* = 47): 38 FIGO I–II, 9 FIGO III–IV; 32 Grade 1, 15 Grade 2–3, 42 N, 5 N+	Hyperplasic endometrial tissue (*n* = 18), Endometrial tissue of healthy cases (*n* = 13)	miR-181a	-	RT-qPCR
Kong et al., 2014 [42]	-	Endometrioid endometrial cancers (*n* = 21)	Endometrial tissue of healthy cases (*n* = 14)	-	miR-30c	RT-qPCR
Jurcevic et al., 2014 [43]	Paraffin	Endometrial cancers (*n* = 30): 10 FIGO I, 10 FIGO II, 10 FIGO III	Endometrial tissue of healthy cases (*n* = 20)	miR-183, -182, 429, -135a, -9-3p, -9, 135b, -200a-5p, -218, -18a-3p	miR-1247, -199b-5p, -214, -370, -424-3p, -376c, -542-5p, -758, -377, 337-5p	RT-qPCR
Tsukamoto et al., 2014 [13]	-	Endometrioid endometrial cancers (*n* = 28): 4N+, 21 N−, 7 FIGO IA Grade 1	Endometrial tissue of healthy cases (*n* = 14)	miR-499, -135b, -205	miR-10b, -195, -30a-5p, -30a-3p, -21	RNAseq RT-qPCR
Xiong et al., 2014 [52]	−80 °C	Endometrioid endometrial cancers (*n* = 15)	Adjacent healthy endometrial tissue (*n* =15)	miR-181c-3p, -25-5p	miR-99a-3p, -96a-5p, -328-3p, -337-3p, let-7c-5p	RNAseq RT-qPCR
Xu et al., 2013 [44]	−80 °C	Endometrioid endometrial cancers (*n* = 71)	Endometrial tissue of healthy cases (*n* = 5) Adjacent healthy endometrial tissue (*n* = 10) Hyperplasic endometrial tissue (*n* = 9)	-	miR-503	RT-qPCR
Torres et al., 2013 [14]	Paraffin −80 °C	Endometrioid endometrial cancers (*n* = 77): 50 FIGO I, 5 FIGO II, 20 FIGO III, 2 FIGO IV 29 Grade 1, 30 Grade 2, 18 Grade 3 29 N+, 15 N−	Endometrial tissue of healthy cases (*n* = 31)	miR-9, -141, -183, -200a, -200a*, -200b, -200b*, -200c, -203, -205, -429, -96, -182, -135b	miR-410	Array RT-qPCR
Torres et al., 2012 [24]	Paraffin −80 °C	Endometrioid endometrial cancers (*n* = 77): 50 FIGO I, 5 FIGO II, 20 FIGO III, 2 FIGO IV 29 Grade 1, 30 Grade 2, 18 Grade 3 29 N+, 15 N−	Endometrial tissue of healthy cases (*n* = 31)	-	miR-99a, -100, -199b	RT-qPCR
Lee et al., 2012 [45]	Paraffin	Endometrial cancers (*n* = 22): 15 FIGO IA, 5 FIGO IB, 2 FIGO IIIC1	Endometrial tissue of healthy cases (*n* = 10) Hyperplasic endometrial tissue (*n* = 21) Atypical hyperplasic endometrial tissue (*n* = 22)	miR-182, -183, -200a, -200c, -205	-	RT-qPCR
Karaayvaz et al., 2012 [23]	Paraffin	Endometrial cancers (*n* = 48): 24 endometrioid, 13 serous, 5 clear cell, 6 others 26 FIGO I, 4 FIGO II, 6 FIGO III, 12 FIGO IV	Adjacent healthy endometrial tissue (*n* = 48)	miR-200c miR-205	-	RT-qPCR
Snowdon et al., 2011 [46]	Paraffin	Endometrioid endometrial cancers (*n* = 19): 9 FIGO IA, 4 FIGO IB, 1 FIGO II.	Endometrial tissue of healthy cases (*n* = 10) Atypical hyperplasic endometrial tissue (*n* = 14)	miR-9/-9*, -18a, -96, -141, -146a, -200a/b/b*/c, -203, -205, -210, -421, -429, -516a-5p, -605, -614, -936	miR-10b*, -23a*, -100, -127-3p, -152, -199b-3p, -199b-5p, -370, 376a/c, -381, -410, -424, -424*, -431, -432, -503, -542-3/5p, -596, 610,630,632, 760	Array RT-qPCR
Cohn et al., 2010 [25]	Paraffin	Endometrial cancers (*n* = 141): 121 endometrioid FIGO I (90 Grade 1, 27 Grade 2, 4 Grade 3), 3 endometrioid FIGO III, 7 serous FIGO III, 4 endometrioid FIGO IV 6 serous FIGO IV	Endometrial tissue of healthy cases: 10 pre-menopausal tissues 10 post-menopausal tissues	miR-9, -19b; -146, -181c, -183, -200c, -205, -223, -423, -425	let-7a, miR-32, -33b, -369, -409, -424, -431, -451, -496, -503, -516	Array RT-qPCR
Ratner et al., 2010 [47]	Paraffin −80 °C	Endometrial cancers (*n* = 90): 57 endometrioid (27 FIGO I, 12 FIGO II, 18 FIGO III), 27 serous 6 carcinosarcoma.	Endometrial tissue of healthy cases (*n* = 5)	miR-182, -183, -200a, -205, -34a, -572, -622, -650	miR-411, -487b	Array RT-qPCR
Chung et al., 2009 [48]	−80 °C	Endometrioid endometrial cancers (*n* = 30): 25 FIGO I–II, 5 FIGO III 19 Grade 1, 11 Grade 2 3 N+, 27 N−	Endometrial tissue of healthy cases (*n* = 22): 7 in proliferating phase 7 in the secretory phase 8 post-menopausal tissues	miR-10a, -17-5p, -23a*, -25, -28, -34a, -95, -103, -106a, -107, -130b, -141, -151, -155, -182, -183, -184, -191, -194, -200a/c, -203, -205, -210, -215, -223, -301, -325, -326, -330	-	RT-qPCR
Wu et al., 2009 [49]	−80 °C	Endometrioid endometrial cancers (*n* = 10): 5 FIGO I, 5 FIGO II	Adjacent healthy endometrial tissue (*n* = 10)	miR-200c, -449, -205, -182, -429, -200b, -96, -31, -141, -200a, -363, -210, -432, -203, -10a, -155, -142-5p	miR-204, -193a, -368, -133b, -193b, -99b	Array RT-qPCR
Boren et al., 2008 [50]	−80 °C	Endometrioid endometrial cancers (*n* = 37)	Endometrial tissue of healthy cases (*n* = 20) Atypical hyperplasic endometrial tissue (*n* = 4)	Let-7c, miR-103, -106a, -107, -181a, -185, -210, -423	let 7i, miR-30c, -152, -193, -221	Puce RT-qPCR

FIGO: International Federation of Gynecology and Obstetrics, N: ganglionic status, RT-qPCR: real-time quantitative polymerase chain reaction.

**Table 2 cancers-11-00832-t002:** Expression profile of microRNAs within neoplastic endometrial tissue according to survival.

Reference	Sample Type	Sample Case	Conclusion
Wang Y. et al., 2018 [16]	-	Endometrial cancers (*n* = 348)	The signature of 6 miRs (miR-15a, miR-142-3p, hsa-miR-142-5P, miR-3170, miR-1976, miR-146a) is associated with a significant decrease in OS (HR = 0.446; 95% CI: 0.218–0.913)
Yan et al., 2018 [17]	Paraffin	Endometrial cancers (*n* = 156): 87 FIGO I, 35 FIGO II, 23 FIGO III, 11 FIGO IV	Increased expression of miR-183-5p is associated with improved prognosis of OS
Deng et al., 2017 [18]	−80 °C	Endometrioid endometrial cancers (*n* = 90)	A decrease in miR-202 expression is associated with a significant decrease in OS (*p* < 0.05)
Tsukamoto et al., 2014 [13]	-	Endometrioid endometrial cancers (*n* = 28): 7 FIGO IA Grade 1, 21 other grades 4 N+, 21 N−	The expression levels of miR-135b, -205, -21, -30a-3p, -499, -10b, -30a-5p, and -195 are not correlated with RFS
Bao et al., 2013 [19]	-	Endometrioid endometrial cancers from Cancer Genome Atlas database (*n* = 279)	Increased expression of miR-204-5p is associated with a nonsignificant improvement in OS (OR = 1.32, *p* = 0.12)
Dong et al., 2013 [20]	Paraffin	Endometrial cancers followed for 15 years (*n* = 32): 15 endometrioid, 8 serous, 5 clear cell, 4 others, -18 FIGO I, 1 FIGO II, 5 FIGO III, 8 FIGO IV	Increased expression of miR-130b is associated with better OS (*p* = 0.05)
Zhang et al., 2013 [21]	Paraffin	Endometrial cancers (*n* = 107): 85 endometrioid, 22 others 30 Grade 1, 39 Grade 2, 16 Grade 3 18 LVSI+, 87 LVSI− 74 FIGO I, 17 FIGO II, 13 FIGO III, 1 FIGO IV 6 N+, 42 N−	The decrease in expression of miR-145 and miR-143 is associated with a nonsignificant decrease in OS (*p* > 0.05)
Torres et al., 2013 [14]	Paraffin −80 °C	Endometrioid endometrial cancers (*n* = 77): 29 Grade 1, 30 Grade 2, 18 Grade 3 15 N+, 29 N−	The expression levels of miR-1228/miR-200c/miR-429 and miR-1228/miR-429 are respectively associated with OS (HR: 2.978, 95% CI: 1.580–5.614, *p* < 0.001) and RFS (HR: 4.149, 95% CI: 2.193–7.852, *p* < 0.001)
Zhai et al., 2013 [22]	Paraffin	Endometrial cancers followed for 15 years (*n* = 32): 15 endometrioid, 8 serous, 5 clear cell, 4 others 17 FIGO I, 1 FIGO II, 5 FIGO III, 9 FIGO IV	Increased expression of miR-194 is associated with better OS (*p* = 0.007)
Karaayvaz et al., 2012 [23]	Paraffin	Endometrial cancers (*n* = 48): 24 endometrioid, 13 serous, 5 clear cell, 6 others 26 FIGO I, 4 FIGO II, 6 FIGO III, 12 FIGO IV	Increased expression of miR-205 is associated with poorer OS (*p* = 0.03) Expression of miR-200c is not correlated with OS (*p* = 0.58)
Torres et al., 2012 [24]	Paraffin −80 °C	Endometrioid endometrial cancers (*n* = 77): 29 Grade 1, 30 Grade 2, 18 Grade 3 50 FIGO I, 5 FIGO II, 15 FIGO III, 2 FIGO IV 29 N+, 15 N−	Increased expression of miR-100 is associated with better OS (*p* = 0.02)
Cohn et al., 2010 [25]	Paraffin	Endometrial cancers (*n* = 141): 128 endometrioid: 121 FIGO I, 3 FIGO III, 4 FIGO IV 13 serous: 7 FIGO III, 6 FIGO IV	Increased expression of miR-199a-5p is associated with better OS (*p* = 0.007) and better RFS (*p* = 0.048)
Hiroki et al., 2010 [26]	−80 °C	Serous adenocarcinoma (*n* = 21): 8 FIGO I, 2 FIGO II, 3 FIGO III, 8 FIGO IV 5 LVSI+, 16 LVSI−	The subexpressions of miR-152, -29b and -455-5p are associated with poorer OS and RFS (*p* < 0.05). Subexpressions of miR-101, -10b *, and -139-5p are associated with poorer OS (*p* < 0.05)
Huang et al., 2009 [27]	-	Endometrial cancers (*n* = 117)	Methylation of the miR-129-2 gene is associated with poorer OS (*p* = 0.039)

FIGO: International Federation of Gynecology and Obstetrics, LVSI: lympho-vascular space involvement, N: ganglionic status, OS: overall survival, RFS: recurrence-free survival, HR: Hazard Ratio.

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
