# Peer review of "The Use of microRNAs in the Management of Endometrial Cancer: A Meta-Analysis"

_cancers, 2019, doi:10.3390/cancers11060832_

Round 1

Reviewer 1 Report

Well written and pertinent.

How do the miRNAs fit into molecular evaluation not just classic anatomic prognostic feature analysis

Author Response

Dear reviewer, Thank you for this very relevant question. For the moment, indeed, there is no integration of the miRs either in the anatomical classification or in the molecular classification. In the future, the miRs could make it possible to better select the patients, notably thanks to the clinical criteria that we develop in the article. We have made your point in our article.

l.152-153: even if there is no integration of the miRs either in the anatomical classification or in the molecular classification until now [2].

Reviewer 2 Report

The manuscript is well written.

Author Response

Dear reviewer, Thank you very much for your comment. We are delighted that you enjoy reading our article.

Reviewer 3 Report

This manuscript was written by  Delangle et. al., title: Use of microRNAs in the Management of Endometrial Cancer: A Meta-Analysis.

The author of this manuscript collects and analyzes 74 original articles and 8 literature reviews published between November 1, 2008 and October 31, 2018

The authors sorted out the related expressions of miRNAs in endometrial cancer. This article is an article with review attributes, but the miRNAs are not enough to explore the pathological molecular mechanisms related to endometrial cancer. 

The authors only sorted and summarized. There is no in-depth discussion on the advantages and disadvantages of miRNA as the biomark of endometrial cancer (current examination method) and irreproducibility.

Author Response

Response to reviewer 3

This manuscript was written by  Delangle et. al., title: Use of microRNAs in the Management of Endometrial Cancer: A Meta-Analysis.

The author of this manuscript collects and analyzes 74 original articles and 8 literature reviews published between November 1, 2008 and October 31, 2018

The authors sorted out the related expressions of miRNAs in endometrial cancer. This article is an article with review attributes, but the miRNAs are not enough to explore the pathological molecular mechanisms related to endometrial cancer.

Dear reviewer, we do agree. It’s one element among others (i.e. anatomic prognostic features, molecular classification) that could help to classify patient in order to better targeting to future treatment.

This warning has been added to the discussion:

153-157: Various studies have focused on miRNAs implication in endometrial cancer mechanisms [58] with no fully established conclusions. Yet, based on new knowledge of miRNAs associated with different prognostic, new pathogenetic classifications for EC may be proposed including miRNAs as one element among others (i.e. anatomic prognostic features, molecular classification) in order to better targeting to future treatment.

The authors only sorted and summarized. There is no in-depth discussion on the advantages and disadvantages of miRNA as the biomark of endometrial cancer (current examination method) and irreproducibility.

Once again, your comment is very relevant. A whole paragraph has been added to detail advantages and disadvantages of miRNAs, especially irreproductibility. This highlights challenges in daily clinical use of miR.

218-232: there are limits to the use of miRs. Indeed, the reproducibility of their detection and quantification is relatively low. This is due to several factors. The accuracy, reproducibility, sensitivity and specificity of the main PCR kits were compared by Mestdagh et al. [83], and present a great disparity in performance. On the other hand, there is currently no consensus regarding the use of housekeeping genes as endogenous internal control when assessing miRNA expression. The housekeeping genes are conventionally selected from a panel of 12 non-coding RNAs (RNU48, RNU44, U75, RNU6B, U6, U54, RNU38B, U18, U49, miR-26b, miR-92a and miR-16) described as stable in tissues or at least used in EC studies [49,84]. Their expression nevertheless shows considerable variability according to some authors [13] and could lead to errors of interpretation and non-reproducibility of the results according to the histological types concerned.

A new technique for the detection and quantification of miRs using Isothermally amplified time-gated Förster resonance energy transfer (TG-FRET), would overcome these drawbacks and already demonstrate its interest in ovarian cancer and breast cancer [85].

Reviewer 4 Report

In this paper the Authors report evidence of miRNAs involvement in endometrial cancer, by analyzing works published between November 1 2008 and October 31 2018. Hence, the Authors suggest the use of certain miRNAs as prognostic factors.

The work is interesting, however there are some concerns.

It would be better to add more discussion on role of the downstream molecules regulated by miRNAs in the endometrial cancer. For example, on Page 9, the Authors mentioned that certain miRNAs “..are involved in the PI3K/Akt/mTOR pathway, most likely through downregulation of the PTEN tumor suppressor”. I suggest that a further discussion on how the PTEN pathway is involved in the endometrial cancer would be beneficial.

The Authors use diverse variants for non coding RNAs (MicroRNAs, miRNAs, MiRs, et al). They should choose one and use only it in the main text and tables.

The manuscript needs careful editing, paying particular attention to spelling. For example, on page 9 line 149, the Authors should write EC, not CE. Moreover, on page 3 line 97 what means p1202 and p1? And EC55, on page 10 line 239?

Please add the reference in the sentence on page 9 line 195-196.

Author Response

Response to reviewer 4

In this paper the Authors report evidence of miRNAs involvement in endometrial cancer, by analyzing works published between November 1 2008 and October 31 2018. Hence, the Authors suggest the use of certain miRNAs as prognostic factors.

The work is interesting, however there are some concerns.

1. It would be better to add more discussion on role of the downstream molecules regulated by miRNAs in the endometrial cancer. For example, on Page 9, the Authors mentioned that certain miRNAs “..are involved in the PI3K/Akt/mTOR pathway, most likely through downregulation of the PTEN tumor suppressor”. I suggest that a further discussion on how the PTEN pathway is involved in the endometrial cancer would be beneficial.

Dear reviewer, thank you for this remark that we found very useful. We fully agree with this one and we have added a sentence recalling the prevalence and the role of gene mutations of the PI3K / AKT / mTOR signaling pathway in endometrial cancer.

l. line 171-180: However, it has been shown that there is a high prevalence of PTEN gene mutation (56-57%) [63,64], PI3K gene mutation (36-39%) [63,64], or even the associated mutation of these two genes (25%) [64]. The loss of PTEN induces a lack of control of Akt phosphorylation, while the mutation of the gene coding for the p85 subunit of PI3K is involved in the overactivation of Akt, both resulting in cell proliferation and resistance to apoptosis [65]. This signaling pathway is also a therapeutic target under study; the use of PI3K / mTOR inhibitor, and the use of temsirolimus as an mTOR inhibitor, would appear to be effective on cells presenting these mutations [66].

2. The Authors use diverse variants for non coding RNAs (MicroRNAs, miRNAs, MiRs, et al). They should choose one and use only it in the main text and tables.

Thank you for your very attentive reading. We have made the necessary corrections. Only the abbreviation "miR" will be used throughout the article, both in the text, subtitles, and tables.

3. The manuscript needs careful editing, paying particular attention to spelling. For example, on page 9 line 149, the Authors should write EC, not CE. Moreover, on page 3 line 97 what means p1202 and p1? And EC55, on page 10 line 239?

As for your previous remark, we thank you for the attention you have had to reading our article. The anomalies that you reported are the consequence of the automatic layout of the bibliography. We apologize. Again, the correct corrections have been made.

4. Please add the reference in the sentence on page 9 line 195-196.

The reference has been added.

73. Navarro F, Lieberman J. miR-34 and p53: New Insights into a Complex Functional Relationship. PLoS ONE. 2015;10(7):e0132767. doi:10.1371/journal.pone.0132767.

Round 2

Reviewer 3 Report

Agree to publish

This manuscript is a resubmission of an earlier submission. The following is a list of the peer review reports and author responses from that submission.